# Chlorate Contamination in Commercial Growth Media as a Source of Phenotypic Heterogeneity within Bacterial Populations

Maxence S. Vincent,[a] Alexandra Vergnes,[a] Benjamin Ezraty[a]

aAix-Marseille Université, CNRS, Laboratoire de Chimie Bactérienne, Institut de Microbiologie de la Méditerranée, Marseille, France

**ABSTRACT** Under anaerobic conditions, chlorate is reduced to chlorite, a cytotoxic compound that triggers oxidative stress within bacterial cultures. We previously found that BD Bacto Casamino Acids were contaminated with chlorate. In this study, we investigated whether chlorate contamination is detectable in other commercial culture media. We provide evidence that in addition to different batches of BD Bacto Casamino Acids, several commercial agar powders are contaminated with chlorate. A direct consequence of this contamination is that, during anaerobic growth, *Escherichia coli* cells activate the expression of *msrP*, a gene encoding periplasmic methionine sulfoxide reductase, which repairs oxidized protein-bound methionine. We further demonstrate that during aerobic growth, progressive oxygen depletion triggers *msrP* expression in a subpopulation of cells due to the presence of chlorate. Hence, we propose that chlorate contamination in commercial growth media is a source of phenotypic heterogeneity within bacterial populations.

**IMPORTANCE** Agar is arguably the most utilized solidifying agent for microbiological media. In this study, we show that agar powders from different suppliers, as well as certain batches of BD Bacto Casamino Acids, contain significant levels of chlorate. We demonstrate that this contamination induces the expression of a methionine sulfoxide reductase, suggesting the presence of intracellular oxidative damage. Our results should alert the microbiology community to a pitfall in the cultivation of microorganisms under laboratory conditions.

**KEYWORDS** agar, Casamino Acids, chlorate, commercial growth media, methionine sulfoxide reductase, oxidative stress, phenotypic heterogeneity

During anaerobic growth, bacterial nitrate reductases (NRs) can reduce chlorate ($ClO_3^-$) to chlorite ($ClO_2^-$) (1–3). While chlorate is stable, the cytotoxicity of chlorite has been recognized for decades (4–6). However, the molecular mechanism underlying this toxicity has only recently been described. Chlorite is harmful to the cell because it oxidizes protein-bound methionine (Met) residues and ultimately provokes protein loss of function (7, 8). In *Escherichia coli*, chlorate reduction activates the two-component signaling system HprSR and induces the expression of the *hiuH-msrPQ* operon (8). MsrP is a periplasmic methionine sulfoxide reductase (Msr) which repairs oxidized protein-bound Met (9). Consequently, the deletion of *msrP* is highly detrimental to anaerobic growth in the presence of chlorate. Under these conditions, *E. coli* colonies exhibit a striking "doughnut-like" morphology in which the interior of the colony is devoid of cells (8). Although the exact mechanism responsible for this macroscopic phenotype remains to be elucidated, this morphological defect can be used as a rapid proxy to detect the presence of chlorate in solid growth media.

In *E. coli*, chlorite production by NRs has been shown to influence periplasmic redox homeostasis. Met residues of the chaperone protein SurA, involved in outer membrane

Address correspondence to Benjamin Ezraty, ezraty@imm.cnrs.fr.

The authors declare no conflict of interest.

protein folding and assembly, are oxidized by chlorite stress, which critically impairs cell survival (8). In *Pseudomonas aeruginosa*, the cytotoxic effect of anaerobiosis-dependent chlorate reduction leads to proteome-wide Met oxidation eventually affecting cell fitness (7). As for *E. coli*, the production of Msr enzymes is essential to rescue damaged proteins and restore cell viability (7). In *Azospira suillum*, chlorite treatment induces the expression of *msrP*, which regenerates sacrificial Met-rich scavengers and thereby decreases the intracellular level of oxidants (10). As such, the protective role of Msr enzymes against the oxidizing effect of chlorite on Met residues is likely to be widely conserved across different bacterial species.

In a previous study, we reported that commercial BD Bacto Casamino Acids (CASA) were contaminated with chlorate (~0.375 mg/g) (8). Consequently, *E. coli* anaerobic growth supplemented with BD Bacto CASA led to critical cellular damage (8). Traditionally, CASA is used as a supplement in microbial growth medium and consists of amino acids and peptides obtained from acid hydrolysis of casein (11). Since CASA is one of the most commonly used growth medium supplements in microbiology, we decided to investigate whether chlorate was detectable in CASA from different commercial origins.

Of the six types of CASA tested in our analysis, only BD Bacto CASA was found to be contaminated with chlorate. Nonetheless, our study revealed that ready-to-use agar powder from numerous suppliers contained significant chlorate levels. Furthermore, we found that chlorate contamination affected bacterial cultures even during aerobic growth. We demonstrated that oxygen depletion after overnight aerobic culture, as is classically performed in most microbiology laboratories, led to the emergence of a subpopulation of cells activating the expression of *msrP*, thus indicating that some cells experienced chlorate/chlorite oxidative stress.

## RESULTS

**Comparison of colony morphology and MsrP production in growth media supplemented with CASA from different suppliers.** Our previous observation showing that BD Bacto CASA contained chlorate (8) prompted us to carry out a systematic comparison of CASA from different suppliers (Fig. 1A). We relied on the colony morphology of a Δ*msrP* strain as a proxy for chlorate contamination (Fig. 1B). As expected, we confirmed that supplementation with BD Bacto Casamino Acids (catalog no. 223050) resulted in aberrant *E. coli* colony morphology after 2 days of anaerobic growth (8) (Fig. 1B). We did not notice any obvious morphological defect for cells plated on solid growth media supplemented with CASA from Merck (catalog no. 2240) or Bio Basic (catalog no. CB3060) or with BD CASA technical grade (catalog no. 223120) or Merck casein hydrolysate (catalog no. 22090) (Fig. 1B). We noted that Δ*msrP* colonies grown on solid growth medium supplemented with BD CASA (catalog no. 228820) displayed a slightly different shape of wild-type (WT) colonies, which could indicate the presence of chlorate traces in these CASA (Fig. 1B).

We further assessed the presence of chlorate by monitoring the expression and the production of MsrP (Fig. 1C and D). In *E. coli*, *msrP* is under the control of the *hiuH* promoter, $P_{hiuH}$ (12), we thus performed $\beta$-galactosidase assays using a strain containing a $P_{hiuH}$-*lacZ* reporter to quantify the expression of *msrP* after overnight anaerobic growth in liquid culture supplemented with the aforementioned CASA. We measured the sensitivity of the reporter using a range of chlorate concentrations and found that it detected up to ~2.2 $\mu$M chlorate (~110 Miller units) (see Fig. S1 in the supplemental material). The reporter activation intensity increased in proportion to increasing doses of chlorate and plateaued at ~20 $\mu$M (Fig. S1). $\beta$-galactosidase activities reflected the colony morphology results and indicated that only BD Bacto CASA triggered the expression of *msrP* (Fig. 1B). Although not significant, levels of $P_{hiuH}$-*lacZ* activation were slightly higher in cultures supplemented with BD CASA (catalog no. 228820) than in cultures not supplemented with CASA (Fig. 1B). Anti-MsrP immunodetection revealed that MsrP is not produced in cultures supplemented with CASA from Merck and Bio Basic and confirmed our previous results (Fig. 1C). We were able to detect

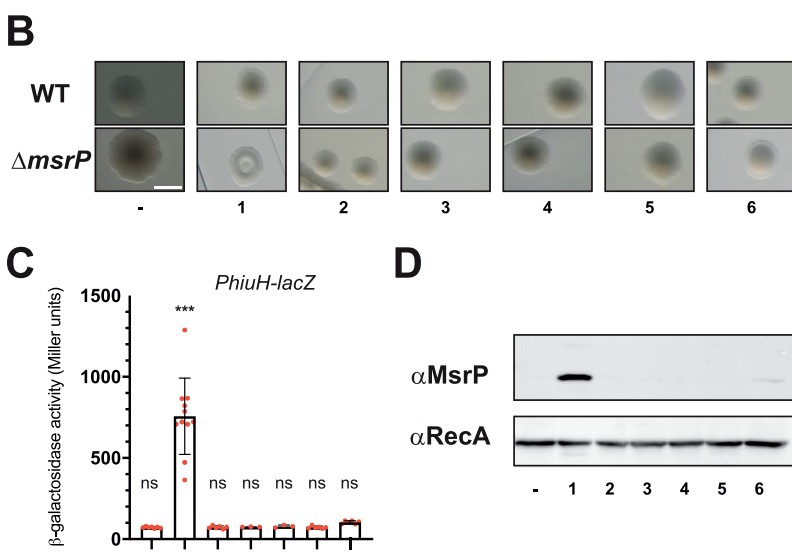

**FIG 1** Supplementation with BD Bacto CASA induces *msrP* under anaerobic conditions. (A) Suppliers, catalog (reference) numbers, and batch numbers of the different CASA powders used in this study. We assigned a number (from 1 to 6) to each CASA reference to facilitate the reading of panels B to D. (B) Colony morphology on LB agarose plates supplemented with different CASA. Wild-type (MG1655) and Δ*msrP* (CH380) strains were streaked onto LB agarose containing 0.4% CASA. The plates were incubated at 37°C under anaerobic conditions for 48 h. Bar = 1 mm. (C and D) After overnight anaerobic growth at 37°C in LB supplemented or not with 0.4% CASA, the expression of the $P_{hiuH}$-*lacZ* reporter (strain CH184) was measured with $\beta$-galactosidase assays (C), and the production of MsrP was monitored by immunoblot analysis (D). (C) Error bars indicate standard deviations ($n = 3$). ns, not significant; \*\*\*, $P < 0.001$ (Dunnett's multiple-comparison test). (D) Anti-RecA was used as a loading control.

weak production of MsrP in cultures supplemented with BD CASA (catalog no. 228820), which strengthened our assumption that this type of CASA may contain traces of chlorate (Fig. 1C).

The production process can lead to variations in the final composition of a growth medium, even for the same brand of CASA. Therefore, we decided to investigate whether the chlorate contamination observed for BD Bacto CASA was restricted to this particular batch (Fig. 2). We tested five different batches of the same item (catalog no. 223050) (Fig. 2A) and found that all of them led to aberrant colony shape, $P_{hiuH}$-*lacZ* activation, and MsrP production (Fig. 2B to D). Although supplementation of growth medium with batch 9313548 led to a moderate colony morphological defect and weak induction of $P_{hiuH}$-*lacZ* (Fig. 2B and C), the production of MsrP was undeniable (Fig. 2D). These results indicated that the different batches of BD Bacto CASA (catalog no. 223050) were likely to be contaminated with chlorate.

**Comparison of colony morphology and MsrP production in growth media supplemented with agar from different suppliers.** Throughout our investigation, we suspected that chlorate contamination might arise from sources other than CASA. Notably, we observed that plating Δ*msrP* cells on LB agar led to the emergence of doughnut-like colonies, which was not the case when cells were plated on LB agarose petri dishes. This observation explains why we used LB agarose instead of LB agar in our previous experiments, assessing the effect of CASA on colony morphology (Fig. 1B

## A

| | Product | Company | Reference | Batch |
|---|---|---|---|---|
| **1** | Bacto Casamino Acids | BD | 223050 | 6266538 |
| **1A** | Bacto Casamino Acids | BD | 223050 | 1280056 |
| **1B** | Bacto Casamino Acids | BD | 223050 | 1082049 |
| **1C** | Bacto Casamino Acids | BD | 223050 | 9313548 |
| **1D** | Bacto Casamino Acids | BD | 223050 | 1300402 |

## B

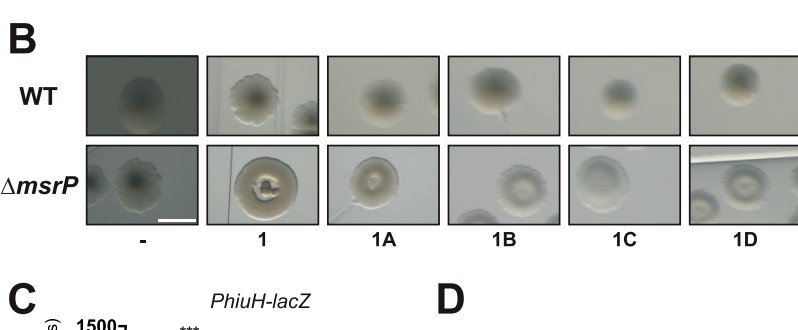

## C

## D

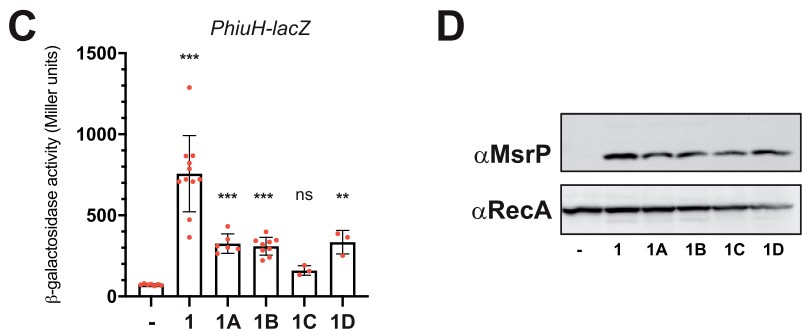

FIG 2 Different batches of BD Bacto CASA induce *msrP* under anaerobic conditions. (A) Batch numbers for the different BD Bacto CASA batches used in this study. We designated the BD Bacto CASA batches as 1A, 1B, 1C, and 1D to facilitate the reading of panels B to D. (B) Colony morphology on LB agarose plates supplemented with different batches of BD Bacto CASA. Wild-type (MG1655) and Δ*msrP* (CH380) strains were streaked onto LB agarose containing 0.4% CASA. The plates were incubated at 37°C under anaerobic condition for 48 h. Bar = 1 mm. (C and D) After overnight anaerobic growth at 37°C in LB supplemented or not with 0.4% BD Bacto CASA, the expression of the P*$_{hiuH}$*-*lacZ* reporter (strain CH184) was measured with β-galactosidase assays (C), and the production of MsrP was monitored by immunoblot analysis (D). (C) Error bars indicate standard deviations ($n = 3$). ns, not significant; **, $P < 0.01$; ***, $P < 0.001$ (Dunnett's multiple-comparison test). (D) Anti-RecA was used as a loading control.

and 2B). Based on this fortuitous observation, we investigated whether chlorate contamination was detected in agar from different suppliers (Fig. 3). Agar from Bio-Rad, BD, Euromedex, and Oxoid (Fig. 3A) all led to an aberrant colony shape, P*$_{hiuH}$*-*lacZ* activation, and MsrP production upon anaerobic growth (Fig. 3B to D). To further confirm that these observations reflected the presence of chlorate, the different agars were analyzed by an external laboratory and chlorate contamination was detected in each sample (Table 1). Importantly, chlorate concentrations found using ionic chromatography were in good agreement with the values obtained with our P*$_{hiuH}$*-*lacZ* strain (Table 1), which suggests that this strain is a robust and rapid biosensor for chlorate detection in the environment.

**Effect of BD CASA supplementation during aerobic growth.** Because NR-mediated reduction of chlorate to chlorite occurs in the absence of oxygen (2, 3, 8), we hypothesized that cells growing under aerobic conditions would not activate *msrP* expression even when cultured in growth media supplemented with chlorate-contaminated compounds. As expected, β-galactosidase activities of P*$_{hiuH}$*-*lacZ* dropped dramatically in the presence of oxygen (Fig. S2). Interestingly, we noticed that, even upon aerobic growth, all β-galactosidase activities were slightly higher when chlorate-contaminated compounds were included in the medium (i.e., CASA or agar) (Fig. S2). We reasoned that this subtle difference might be due to the uneven oxygenation of the culture or cell-to-cell variability in oxygen accessibility leading to the activation of P*$_{hiuH}$*-*lacZ* in a few cells of the population.

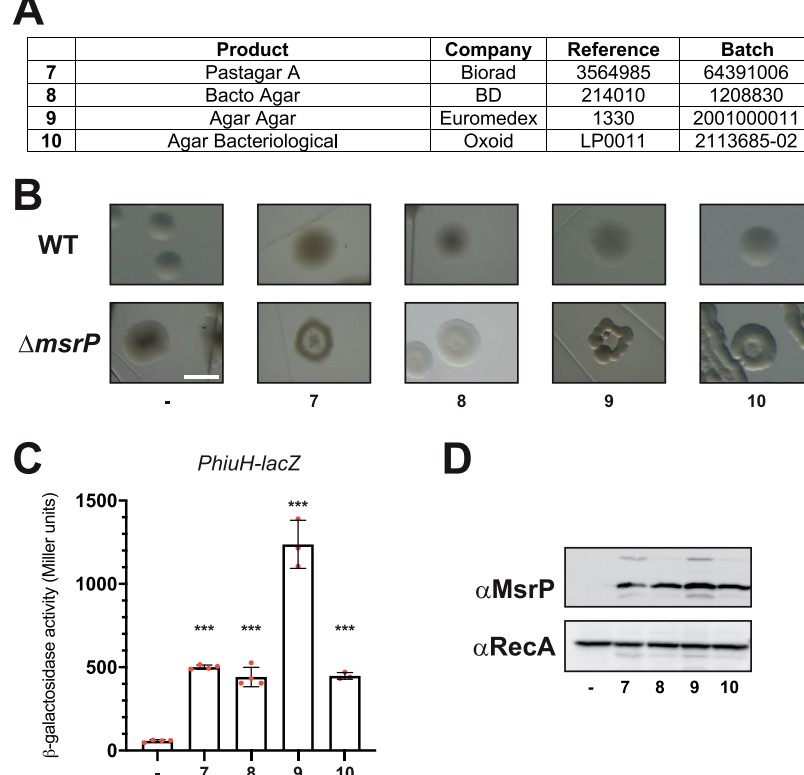

**FIG 3** Agar from different suppliers induces *msrP* under anaerobic conditions. (A) Suppliers, catalog (reference) numbers, and batch numbers of the different agar powders used in this study. We assigned a number (from 7 to 10) to each agar reference to facilitate the reading of panels B to D. (B) Colony morphology of wild-type (MG1655) and Δ*msrP* (CH380) strains streaked onto LB agarose (−) or LB agar. The plates were incubated at 37°C under anaerobic conditions for 48 h. Bar = 1 mm. (C and D) After overnight anaerobic growth at 37°C in LB supplemented or not with agar, the expression of the $P_{hiuH}$-*lacZ* reporter (strain CH184) was measured using β-galactosidase assays (C), and the production of MsrP was monitored by immunoblot analysis (D). (C) Error bars indicate standard deviations ($n = 3$). \*\*\*, $P < 0.001$ (Dunnett's multiple-comparison test). (D) Anti-RecA was used as a loading control.

To test this hypothesis, we performed single-cell imaging analyses of $P_{hiuH}$ activation during aerobic growth in the presence of BD Bacto CASA (Fig. 4). We performed batch culture growth in 50-mL flasks (Erlenmeyer) or 15-mL tubes (Falcon) filled with different volumes of LB supplemented or not with BD Bacto CASA. For the 50-mL-flask condition, we used 5, 10, and 20 mL of growth medium (which corresponded to 1/10, 1/5, and 1/2.5 ratios of culture volume [$V_{culture}$] to flask volume [$V_{flask}$], respectively). For the 15-mL-tube condition, we used 0.5, 1, and 5 mL of growth medium (which corresponded to 1/30, 1/15, and 1/3 $V_{culture}/V_{tube}$ ratios, respectively). Relying on a plasmid-borne $P_{hiuH}$-GFP (green fluorescent protein) reporter (13), we quantified the mean single-cell fluorescence after aerobic growth overnight (~14 h) under these different conditions (Fig. 4A to D). Overall, the fluorescence of cells containing the $P_{hiuH}$-GFP reporter was higher than that of the WT MG1655 strain, which indicated that $P_{hiuH}$ was activated at a basal level regardless of the presence of CASA in the culture (Fig. 4A to D). Most importantly, cultures grown in medium supplemented with CASA exhibited a wide variability of single-cell fluorescence (Fig. 4A to D). $P_{hiuH}$-GFP intensities ranged from 1- to 7-fold for the 1/5 and 1/2.5 ratios in flasks and the 1/3 ratio in tubes (Fig. 4C and D). In order to obtain a better estimate of the population heterogeneity, we calculated the coefficient of variation (CV), which is commonly used to measure the dispersion of a probability distribution.

The CV calculation indicated that the addition of CASA to cultures grown aerobically in flasks significantly increased the dispersion of $P_{hiuH}$-GFP values in the population for all $V_{culture}/V_{flask}$ ratios tested (Fig. 4E). We noticed that the fraction of $P_{hiuH}$-GFP-expressing cells

**TABLE 1** Chlorate detection in agar

| Item no.[a] | Agar supplier | Catalog no. | Chlorate concentration ($\mu$M) | |
|---|---|---|---|---|
| | | | Ionic chromatography | CH184 biosensor |
| 7 | Bio-Rad | 3564985 | 19.9 | 7.6 |
| 8 | BD | 214010 | 15.9 | 6.9 |
| 9 | Euromedex | 1330 | 14.9 | 14.8 |
| 10 | Oxoid | LP0011 | 15.9 | 7 |

[a]Number assigned for the purposes of this study (see the figures).

slightly increased in larger culture volumes even in the absence of CASA (Fig. 4E). Although we have no satisfactory explanation for this effect, we cannot rule out the possibility that (i) some other compounds in our growth media contained traces of chlorate (e.g., sterile $H_2O$ or the flask itself after being sterilized), (ii) cell-to-cell variation in plasmid copy number is affected by the $V_{culture}/V_{flask}$ ratio, or (iii) part of the expression of *msrP* detected in our analysis is affected by the $V_{culture}/V_{flask}$ ratios irrespectively of the presence of chlorate (e.g., nutrient starvation). This effect was not observed for cultures grown in tubes (Fig. 4D to F), which supported the hypothesis of residual chlorate contamination of laboratory glassware. Most importantly, we detected the activation of *msrP* for the 1/15 and 1/3 $V_{culture}/V_{tube}$ ratios only when cultures were supplemented with CASA (Fig. 4D to F). These results indicated that during aerobic growth, the addition of chlorate-contaminated compounds to bacterial cultures increased phenotypic heterogeneity within isogenic cell populations.

## DISCUSSION

Our investigation revealed that chlorate contamination is essentially restricted to one brand of CASA (BD Bacto Casamino Acids). However, we determined that several agars contained chlorate. Since agar is arguably the most commonly used solidifying agent for microbiological media, its contamination with chlorate could have important consequences for the growth of bacterial cultures. Moreover, because the chlorate contamination was found in agar powders from different suppliers, it is unlikely to be a by-product of the production process.

Where does chlorate contamination in agar come from? Agar is collected from agarophyte seaweed, and the extraction process involves multiple rounds of washing with water (14). During these steps, manufacturers add sodium hypochlorite (bleach) and other chlorine chemicals for decontamination and decolorization (15). It is thus tempting to speculate that this procedure is the source of chlorate contamination in agar. Furthermore, agarose preparation involves additional purification steps that remove residual chlorate traces, which could explain the absence of $\Delta msrP$ doughnut-like colonies and MsrP production under anaerobic conditions using LB agarose instead of LB agar.

Whether the composition of agar causes different phenotypes has been debated in several studies: for instance, it is known that agar composition affects swarming motility (16) and killing mediated by type VI secretion systems (17). Recently, it was proposed that reactive oxygen species found in agar were responsible for growth inhibition of environmental microbes (18). This observation is particularly interesting given that oxidative levels vary with the commercial origin of microbial growth media (19) and with storage conditions (20). In this respect, our study could unveil the source of yet-unexplainable phenotypes or the difficulties in reproducing significant results.

Finally, although the reduction of chlorate to chlorite is mediated by anaerobic respiratory complexes, our investigation demonstrated that the influence of chlorate contamination must be questioned even for cultures grown under conditions usually considered aerobic. We propose that partial oxygen depletion triggers cell-to-cell variability in response to chlorate stress. A decrease in oxygen levels under aerobic growth conditions was previously suggested to induce phenotypic changes in clonal populations (21). Furthermore, it is known that growth medium variability partly explains inherent phenotypic fluctuations among isogenic bacterial cells (22). Therefore, it is likely

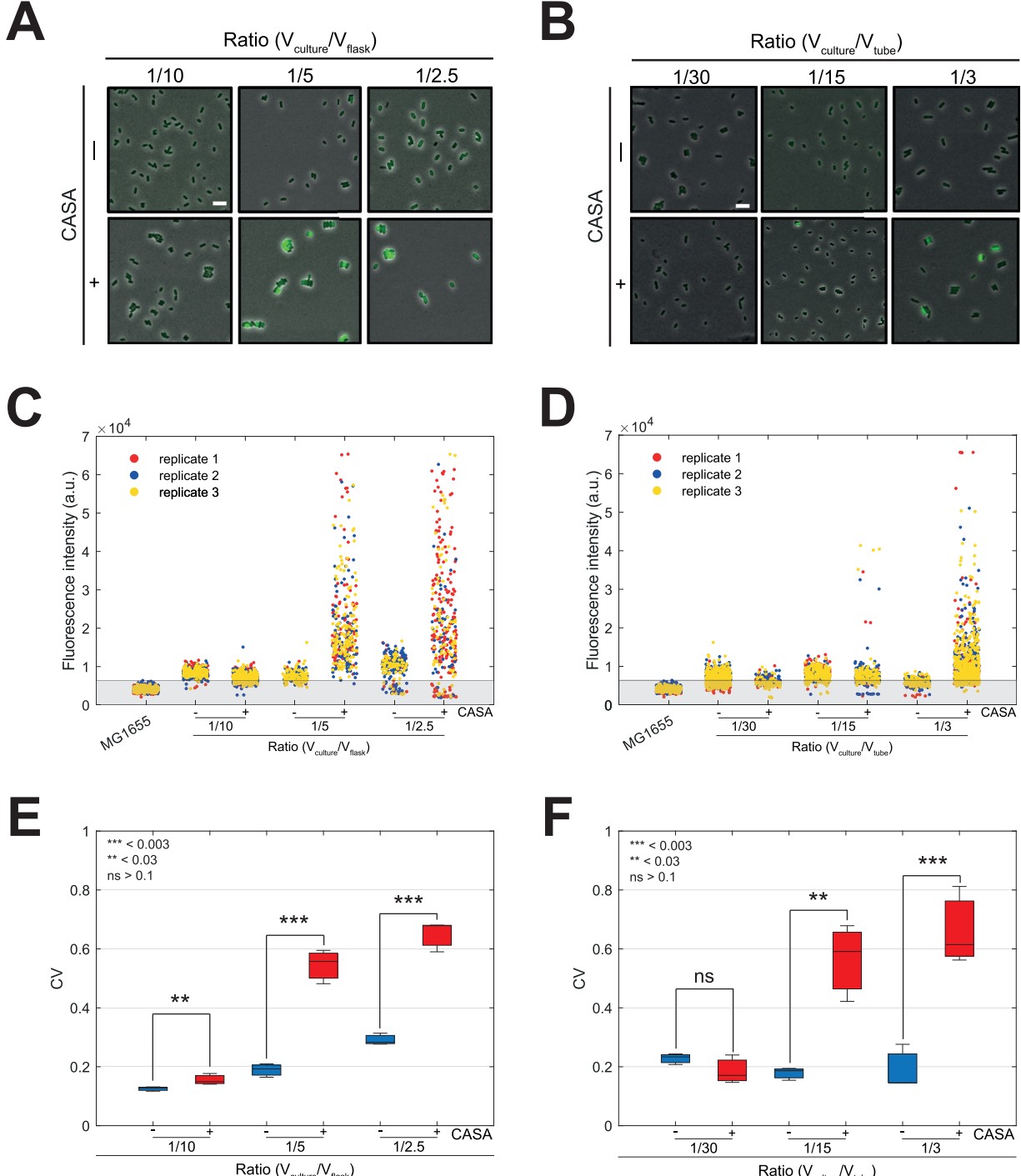

**FIG 4** Chlorate contamination influences population heterogeneity during aerobic growth. (A and B) Examples of single-cell imaging snapshots. Cells transformed with a plasmid carrying the $P_{hiuH}$-*gfpmut2* fusion were imaged after overnight growth in flasks (A) or tubes (B) filled with different volumes of growth medium supplemented or not with 0.4% BD Bacto CASA. Phase-contrast images were merged with their respective fluorescent channel (GFP) images. The fluorescence level was normalized to the image displaying the highest fluorescence level so that fluorescence intensities could be compared between all images. Bar = 1 $\mu$m. (C and D) Each dot represents the mean single-cell fluorescence level in flasks (C) or tubes (D). Three replicates are represented by different colors. The shaded area represents the background fluorescence and is defined based on the maximal fluorescence intensity value detected for WT MG1655 cells. (E and F) Box plots of the coefficient of variation for the different replicates shown in panels C and D. *t* tests were used for statistical analyses.

**TABLE 2** Strains and plasmid used in this study

| Strain or plasmid | Genotype | Source or reference |
|---|---|---|
| Strains | | |
| MG1655 | WT | Laboratory collection |
| CH184 | PM1205 *hiuH-lacZ* | 12 |
| CH380 | MG1655 Δ*msrP* | 8 |
| MV19 | MG1655 carrying plasmid pMV10 | This study |
| | | |
| Plasmid | | |
| pMV10 | pUA66 carrying $P_{hiuH}$::*gfpmut2* | 13 |

that chlorate contamination could act as a source of phenotypic heterogeneity within bacterial populations.

## MATERIALS AND METHODS

**Chemicals reagents, strains and plasmids.** Suppliers, catalog numbers, and batch numbers for agars and CASA used in this study are given in Fig. 1 to 3. Sodium chlorate, which was used to determine biosensor sensitivity, was purchased from Acros Organics (catalog no. 223222500). Antibiotics were used at the following concentrations: ampicillin, 50 $\mu$g/mL, and kanamycin, 25 $\mu$g/mL. The strains and plasmid used in this study are given in Table 2.

**Colony morphology assays.** Cells were streaked onto LB agar or LB agarose (Sigma A9509) plates supplemented or not with CASA (0.4% [wt/vol]). The plates were incubated at 37°C under anaerobic conditions for 48 h. Growth under anaerobiosis was carried out using the GENbox Anaer generator (bioMérieux) in a dedicated chamber. The homogeneity of the colony shape across the plate was verified, and the plates were scanned using a CanoScan 4200F scanner. Single colonies were imaged using a binocular magnifier (Nikon SMZ800N).

**$\beta$-Galactosidase assays.** Construction of the $P_{hiuH}$-*lacZ*-containing strain (CH184) is described in reference 12. CH184 was grown overnight at 37°C under aerobic or anaerobic conditions in LB supplemented or not with CASA (0.4% [wt/vol]) or agar. Activities of $\beta$-galactosidase were measured as previously described (23). Growth under anaerobiosis was carried out using 2-mL tubes full to the brim.

**Immunoblot analysis of MsrP production.** CH184 cultures were grown overnight anaerobically at 37°C in 2-mL tubes full to the brim. Cells were harvested, and the pellets were suspended in Laemmli buffer (2% SDS, 10% glycerol, 60 mM Tris-HCl [pH 7.4], 0.01% bromophenol blue). The amount of protein loaded onto the gel was standardized for each culture based on its $A_{600}$ value. Samples were then heated for 10 min at 95°C and separated by SDS-PAGE. Immunoblot analysis was performed according to standard procedures: the primary antibody was a guinea pig anti-MsrP antibody (kindly provided by Jean-François Collet, De Duve Institute, Belgium). The secondary antibody was an anti-guinea pig IgG conjugated to horseradish peroxidase (HRP) (Promega). For loading controls, a rabbit polyclonal anti-RecA antibody was used (Abcam 63797). Chemiluminescence of immunoblots was measured with an ImageQuant LAS4000 camera (GE Healthcare Life Sciences).

**Chlorate detection in agar.** Agar powders were analyzed by Flandres Analyses (www.flandres-analyses.fr) using liquid-phase ion chromatography according to the AFNOR NF EN ISO 10304-1 protocol.

**Single-cell imaging of *msrP* induction and image analysis.** The MG1655 strain was transformed with a low-copy-number plasmid (SC101 origin) carrying the $P_{hiuH}$ promoter sequence upstream of the coding sequence for *gfpmut2* (green fluorescent protein) (13). Cells were grown in either 15-mL tubes (Falcon) or 50-mL flasks (Erlenmeyer) overnight (approximately 14 h) using different volumes of growth media (0.5, 1, and 5 mL for growth in tubes and 5, 10, and 20 mL for growth in flasks) supplemented or not with 0.4% BD Bacto CASA. One microliter of cells was collected, spotted onto 1% agarose (Bio-Rad no. 1613100) pads, and imaged at ×100 (numerical aperture [NA] 1.45 objective) using a Nikon Ti-E inverted fluorescence microscope equipped with a complementary metal oxide semiconductor (CMOS) camera (Hamamatsu Orca Fusion). Single-cell outlines were automatically segmented from phase-contrast images using a modified version of MicrobeTracker (24) combined with SuperSegger (25). The $P_{hiuH}$-GFP intensity per cell was measured from the average pixel intensity within each segmented cell using a custom-written MATLAB (MathWorks) script. For each replicate, the coefficient of variation was calculated as $\sigma/\mu$, where $\sigma$ is the standard deviation and $\mu$ is the mean of the single-cell fluorescence level in the population.

## SUPPLEMENTAL MATERIAL

Supplemental material is available online only.
**SUPPLEMENTAL FILE 1**, PDF file, 0.1 MB.

## ACKNOWLEDGMENTS

We thank members of the Ezraty group for insightful discussions and are particularly grateful to Laurent Loiseau for his help.

This work was supported by grants from the Agence Nationale de la Recherche (ANR) (ANR-21-CE15-0039 NeutrOX), the Prematuration Program of the CNRS (2021-22-Bachlosens), and SATT Sud-Est (2023-Maturation/Bachlosens).

We declare no conflict of interest.

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
