## [Reviewer comments · Microbiology Spectrum]

Microbiology Spectrum

Chlorate contamination in commercial growth media as a source of phenotypic heterogeneity within bacterial populations

Maxence Vincent, Alexandra Vergnes, and Benjamin Ezraty

Corresponding Author(s): Benjamin Ezraty, CNRS Delegation Provence et Corse

Review Timeline:

Submission Date:	December 5, 2022
Editorial Decision:	December 29, 2022
Revision Received:	January 6, 2023
Accepted:	January 18, 2023

Editor: Blaire Steven

Reviewer(s): Disclosure of reviewer identity is with reference to reviewer comments included in decision letter(s). The following individuals involved in review of your submission have agreed to reveal their identity: Ravikumar R Patel (Reviewer #1)

Transaction Report:

DOI: <https://doi.org/10.1128/spectrum.04991-22>

December 29, 2022

Dr. Benjamin Ezraty
CNRS Delegation Provence et Corse
Marseille
France

Re: Spectrum04991-22 (Chlorate contamination in commercial growth media as a source of phenotypic heterogeneity within bacterial populations)

Dear Dr. Benjamin Ezraty:

After review by a referee and myself, we would be happy to accept the manuscript for publication after a few minor edits listed below. Please follow the instructions indicated and re-submit the manuscript.

Thank you for your interest in publishing with Spectrum.

Thank you for submitting your manuscript to Microbiology Spectrum. As you will see your paper is very close to acceptance. Please modify the manuscript along the lines I have recommended. As these revisions are quite minor, I expect that you should be able to turn in the revised paper in less than 30 days, if not sooner. If your manuscript was reviewed, you will find the reviewers' comments below.

When submitting the revised version of your paper, please provide (1) point-by-point responses to the issues raised by the reviewers as file type "Response to Reviewers," not in your cover letter, and (2) a PDF file that indicates the changes from the original submission (by highlighting or underlining the changes) as file type "Marked Up Manuscript - For Review Only". Please use this link to submit your revised manuscript. Detailed instructions on submitting your revised paper are below.

Link Not Available

Sincerely,

Blaire Steven

Reviewer comments:

Reviewer #1 (Comments for the Author):

This study explored the microbial medium, Bacto Casamino Acid, of different companies for chlorate contamination. First, the authors proved that due to chlorate contamination in the CASA, *E. coli* activates the expression of *msrP* in an aerobic condition. Furthermore, they compared the expression level of *msrP* in different growth media and even in different batches of BD Bacto Casamino Acid. Finally, they showed that different batches originating from BD were contaminated with chlorate. Furthermore, the authors reported that agar from different makes was also infected with chlorate. This study is essential for the microbiology community of those studying bacterial heterogeneity in the lab. This study is well-designed, but the manuscript needs minor modifications before being accepted for publication.

Can the authors provide more detail about the *Msr* enzyme, its function in another group of bacteria, and whether this enzyme is involved in protecting other genera of bacteria against chlorite toxicity?

The section "importance" of the manuscript is very similar to the abstract, and the authors consider modifying this section and mention here only the importance of the study.

Abstract line no. 14 "e. This study investigates whether chlorate 15 contamination is detectable in other commercial culture media." Grammatically incorrect; consider rewriting this sentence.

In lines 281 to 283, the authors mentioned the PhiH-GFP intensity per cell was measured from the average pixel intensity. However, the authors should have explained how they measured the intensity and what software they used to measure the cell intensity. Consider providing details.

The authors mistakenly wrote the title conclusion instead of the title discussion and did not provide a conclusion; consider correcting this.

The short form of the liter should be capitalized "L," and not a lowercase "l" consider correcting throughout the manuscript.

There should be a space between the temperature number and the degree of celsius; consider correcting.

References in the reference list are not uniformly formatted, for example references no. 12, 17, 18, and 23.

Preparing Revision Guidelines

Please return the manuscript within 60 days; if you cannot complete the modification within this time period, please contact me. If you do not wish to modify the manuscript and prefer to submit it to another journal, please notify me of your decision immediately so that the manuscript may be formally withdrawn from consideration by Microbiology Spectrum.

Reviewer #1 (Comments for the Author):

This study explored the microbial medium, Bacto Casamino Acid, of different companies for chlorate contamination. First, the authors proved that due to chlorate contamination in the CASA, *E. coli* activates the expression of *msrP* in an aerobic condition. Furthermore, they compared the expression level of *msrP* in different growth media and even in different batches of BD Bacto Casamino Acid. Finally, they showed that different batches originating from BD were contaminated with chlorate. Furthermore, the authors reported that agar from different makes was also infected with chlorate. This study is essential for the microbiology community of those studying bacterial heterogeneity in the lab. This study is well-designed, but the manuscript needs minor modifications before being accepted for publication.

Can the authors provide more detail about the Msr enzyme, its function in another group of bacteria, and whether this enzyme is involved in protecting other genera of bacteria against chlorite toxicity?

We would like to thank the reviewer for this suggestion. We have added several sentences to the introduction of the manuscript:

L57-60: "In Azospira suillum, chlorite treatment induces the expression of msrP that regenerates sacrificial Met-rich scavengers and thereby decreases the intracellular level of oxidants (10). As such, the protecting role of Msr enzymes against the oxidizing effect of chlorite on Met residues is likely to be widely conserved across different bacterial species."

The section "importance" of the manuscript is very similar to the abstract, and the authors consider modifying this section and mention here only the importance of the study.

We thank the reviewer for mentioning this point. We have removed modified the section "importance" and removed unnecessary sentences.

Abstract line no. 14 " This study investigates whether chlorate 15 contamination is detectable in other commercial culture media." Grammatically incorrect; consider rewriting this sentence.

Thank you. We have made the following alteration to the manuscript:

L14: "In this study, we investigate whether chlorate contamination is detectable in other commercial culture media."

In lines 281 to 283, the authors mentioned the Phi_uH-GFP intensity per cell was measured from the average pixel intensity. However, the authors should have explained how they measured the intensity and what software they used to measure the cell intensity. Consider providing details.

Thank you. We have made the following alteration to the manuscript:

L277-278: "The Phi_uH-GFP intensity per cell was measured from the average pixel intensity within each segmented cell using a custom-written MATLAB (Mathworks) script."

The authors mistakenly wrote the title conclusion instead of the title discussion and did not provide a conclusion; consider correcting this.

Thank you. This has been changed in the revised manuscript.

The short form of the liter should be capitalized "L," and not a lowercase "l" consider correcting throughout the manuscript.

Thank you. We have corrected this throughout.

There should be a space between the temperature number and the degree of celsius; consider correcting.

Thank you. We have corrected this throughout.

References in the reference list are not uniformly formatted, for example references no. 12, 17, 18, and 23.

Thank you. References have been consistently formatted.

January 18, 2023

Dr. Benjamin Ezraty
CNRS Delegation Provence et Corse
Marseille
France

Re: Spectrum04991-22R1 (Chlorate contamination in commercial growth media as a source of phenotypic heterogeneity within bacterial populations)

Dear Dr. Benjamin Ezraty:

Your manuscript has been accepted, and I am forwarding it to the ASM Journals Department for publication. You will be notified when your proofs are ready to be viewed.

Sincerely,

Blaire Steven
Editor, Microbiology Spectrum
